# A Diagonal Structured State Space Model on Loihi 2 for Efficient Streaming Sequence Processing

## Abstract

The unsustainable rise in energy cost from increasingly capable deep learning systems spurs computer architecture innovation beyond conventional deep learning accelerators such as GPUs. However, a novel computer architecture presents a problem: much of deep learning research has been optimized for conventional computer architectures, and the extent to which modern deep learning models can unlock improved efficiency on a novel computer architecture is not well understood. In this work, we demonstrate for the first time that a State Space Model (SSM) can achieve substantial efficiency improvement when mapped to Loihi 2, a state-of-the-art neuromorphic research chip, versus a Jetson Orin Nano GPU (Jetson). Specifically, we benchmark our SSM on sMNIST, psMNIST, and sCIFAR online token-by-token inference and find approximately 1000x increased energy efficiency and 75x improved latency and throughput on Loihi 2 with a decrease in accuracy of less than one to three percentage points compared to the full precision implementation on Jetson. We comprehensively tailor our implementation to Loihi-specific features and constraints, such as the co-location of memory and compute as well as fixed precision arithmetic. Our results elucidate how SSMs meaningfully bridge conventional and neuromorphic hardware via their dual nature: SSMs can operate in an offline mode using convolution or scan, which is efficient on a GPU, or in an online mode as a recurrent network, which we show is efficient on Loihi 2. This work provides a foundation for performant sequence models on neuromorphic hardware, potentially unlocking substantial improvements in latency-sensitive or energy-limited online inference applications, such as speech enhancement or vision for robotic control.

## 1 Introduction

Deep learning systems exhibit improved representational power and AI capabilities as their computational cost increases, and their commensurate rising energy use has been driving unprecedented innovation in computer architecture. The growth of compute (FLOPs) and memory (bandwidth) requirements of data-center-scale deep learning systems such as Large Language Models (LLMs) vastly outpaces the compute and memory delivered by year-over-year improvements in GPUs, the standard workhorse computer architecture of deep learning (Gholami et al., 2024). Similarly, at the edge, the proliferation of intelligent Internet of Things (IoT) devices pushes demand for increasingly capable deep learning systems under power, latency, privacy, and connectivity constraints (Mao et al., 2024; Meuser et al., 2024). To deliver deep learning training and inference efficiency improvements beyond what GPU architectures can offer, in recent years we have seen a "Cambrian explosion" of new computer architectures (Sukumar et al., 2021), such as the TPU (Jouppi et al., 2017), the Cerebras WSE-2 (Lie, 2024), neuromorphic chips such as Loihi 2 (Labs, 2021) or DYNAP-SE2 (Richter et al., 2024), and even analog AI chips, (e.g., Ambrogio et al., 2023), to name a few.

This Cambrian explosion, however, faces a problem known as the hardware lottery: novel computer architectures struggle to take hold because years of deep learning research has targeted GPUs (Hooker, 2021). The continual investment in GPU-focused algorithms has certainly created incredible GPU-based deep learning systems such as ChatGPT (Achiam et al., 2023). Concurrently,

however, deep learning research has become locked-in to GPU implementation at some level, as the most successful algorithmic innovations have compounded around the GPU architecture. The extent to which today's most impactful deep learning technologies can be transferred to novel computer architectures for improved efficiency remains unclear.

In this paper, we show a positive example of broad relevance for how one can substantively improve a modern deep learning system's efficiency on a highly-differentiated novel computer architecture. In particular, we map a State Space Model (SSM) to Loihi 2, a state-of-the-art neuromorphic research chip (Labs, 2021). SSMs are efficient sequence models that rival transformers (Gu et al., 2021b). Importantly, SSMs can perform inference in a convolution or scan mode which is efficient on GPUs, and in a recurrent online token-by-token processing mode. The recurrent formulation of SSMs with their local stateful computation aligns well with the architecture of neuromorphic processors, in which compute and memory are co-located (Davies et al., 2021). This is in contrast to GPUs, where the separation of compute and memory tends to provide efficiency only for batched, predictable, or highly structured computations and memory accesses, such as convolutions (Kumar, 2023). We show that this online token-by-token recurrent mode is in fact extremely efficient on the Loihi 2 architecture. Importantly, online token-by-token inference is highly salient for a wide variety of latency-sensitive or energy-constraint applications such as robotics, autonomous vehicles, and speech enhancement.

Our main contributions are as follows:

1. We demonstrate for the first time an SSM that runs on neuromorphic hardware.

2. We present our Post Training Quantization (PTQ) and Quantization Aware Fine Tuning (QAFT) techniques underpinning the successful mapping of our SSM to Loihi 2.

3. We benchmark our SSM's **online token-by-token** sMNIST, psMNIST, and sCIFAR inference on Loihi 2 versus a recurrent SSM baseline on an edge GPU, Jetson Orin Nano, and we find approximately 1000x improved energy efficiency, 75x decreased latency, and 75x increased throughput, with only a modest decrease in classification accuracy.

4. We also benchmark our SSM's **offline sample-by-sample** sMNIST, psMNIST, and sCIFAR inference on Loihi 2 versus a convolutional SSM baseline on Jetson Orin Nano, and we find Jetson Orin Nano to be advantageous in this context, especially when using batching. This result helps elucidate a more comprehensive account of the differing scenarios for which neuromorphic versus GPU architectures are preferable.

## 2 BACKGROUND

### 2.1 PRELIMINARIES ON NEUROMORPHIC COMPUTING AND LOIHI 2

Neuromorphic computing draws inspiration from the brain's highly efficient approach to information processing. Despite operating at around 20 watts of power, the brain executes complex tasks that include perception, decision-making, coordination, and learning—all in real-time. Neuromorphic computers aim to emulate the brain's incredible efficiency by incorporating the pertinent computational paradigms of the brain's architecture: highly parallel processing, event-driven computation, memory-compute co-location, inherent scalability, and stochasticity (for a review see Schuman et al., 2022). The highly parallel processing and memory-compute co-location help address the aforementioned growing compute and memory interface shortcomings of GPU architectures (Gholami et al., 2024). Furthermore, event-driven computation promotes energy efficiency, as computations and communications are only performed when necessary.

The digital neuromorphic processor Loihi 2 (Labs, 2021) realizes the principles of neuromorphic computing throughout its architecture. Loihi 2 is comprised of computational units, called neuro cores, that contain programmable neurons which communicate by sending spiking events through a Network-on-Chip mesh. These spiking events are small message packets, which carry either a binary or integer payload; spiking events with integer payload are referred to as *graded spikes*. With co-located memory, the neuro cores enable various types of synaptic connectivity, including linear projections and convolutions, as well as more flexible patterns like (pseudo) stochastic or factorized connections. Importantly, Loihi 2 allows users to define custom stateful neurons in the neuro cores

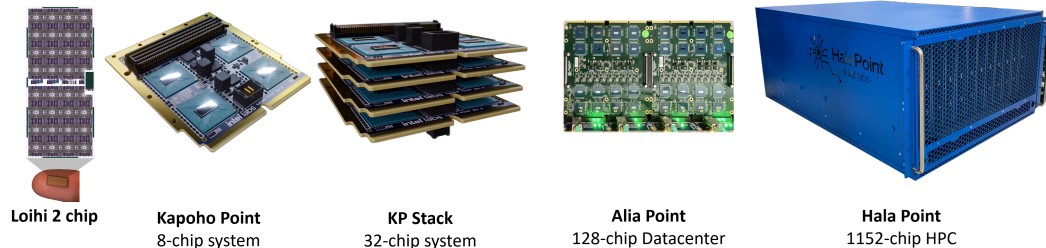

**Loihi 2 chip**

**Kapoho Point**
8-chip system

**KP Stack**
32-chip system

**Alia Point**
128-chip Datacenter

**Hala Point**
1152-chip HPC

Figure 1: Different form factors of Loihi 2 chips, from 31 mm$^2$ single chip to datacenter scale systems. Each Loihi 2 chip features 120 neuro cores dedicated to executing neuromorphic workloads, along with six embedded processor cores for managment. The Loihi 2 chip also includes a dedicated spike I/O unit with a 10 Gbps Ethernet interface. Loihi 2 chips can be connected through six asynchronous parallel interfaces, enabling the extension of the neuromorphic mesh in three dimensions.

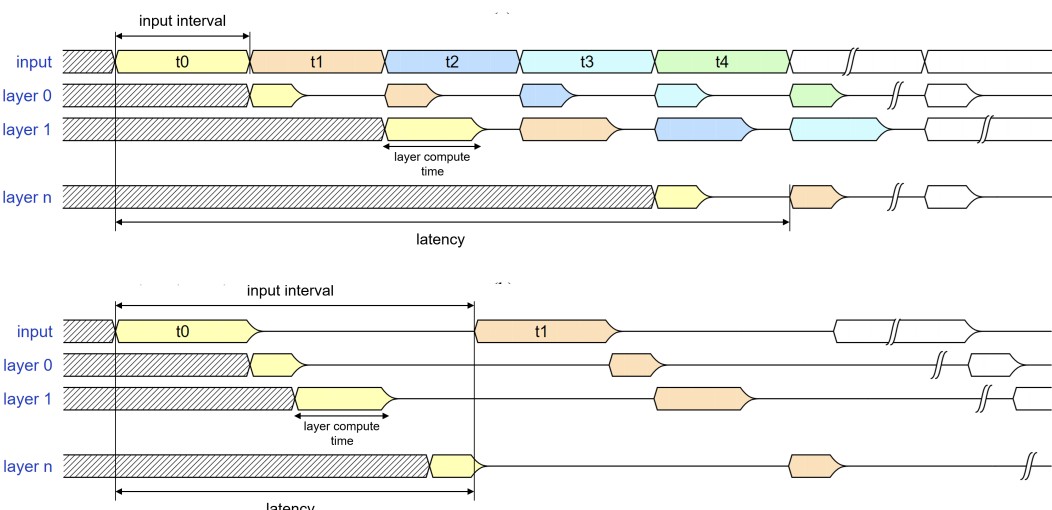

Figure 2: Different modes of inference on Loihi 2. (a) Pipelined execution mode prioritizes throughput at the cost of increased latency. (b) Fall-through mode prioritizes latency at the cost of decreased throughput.

using microcode with a flexible instruction set, including multiplication, (saturated) addition, comparisons, jumps, and bit shifts. Additionally, Loihi 2 is inherently scalable (see Figure 1), equipped with the necessary infrastructure for low-latency interchip and external interface event-based spike communication. The merit of the Loihi 2 architecture has been demonstrated in a variety of domains, including model predictive control (Mangalore et al., 2024), solving QUBO problems (Pierro et al., 2024), monocular depth estimation (Chiavazza et al., 2023), and efficient video and audio processing (Shrestha et al., 2024).

Furthermore, in contrast to most computer chips which use a synchronous clock, Loihi 2 is an *asynchronous* system, which affords energy efficiency and a flexible latency-throughput trade-off. A system of Loihi 2 chips, no matter whether it is single or thousand chips, operates asynchronously performing only the necessary computations as quickly as possible and synchronizing the advance of time-step via a barrier-synchronization mechanism. When considering a deep neural network, the asynchronous nature of Loihi 2 also allows us to seek a sweet spot in latency-throughput trade-off as depicted in Figure 2: one can execute workloads on Loihi 2 scheduling inputs as fast as possible to maximize throughput in a *pipelined* manner, where all of the layers of the network are active in every time-step or allow the current input to propagate through all the layers in the network before injecting the next input in a *fall-through* fashion so that each layer spends a minimum amount of time necessary, thus minimizing the overall latency of the inference per input. Importantly, the distinction between pipelined execution and fall-through execution is not binary but can be considered

a continuum, where many latency-throughput trade-off performance points can be achieved. For example, one could opt for the amount of pipelining that provides the minimal latency under the condition that Loihi 2's throughput can keep up with the sampling rate of an input stream.

## 2.2 DEEP STATE-SPACE MODELS

Recently, a family of linear recurrent architectures, deep SSMs, has emerged. Deep SSMs, such as S4, S4D, Liquid-S4, S5 and Mamba (Gu et al., 2021a; Smith et al., 2022; Hasani et al., 2022; Gu et al., 2022; Gu & Dao, 2023), are based on the memory property of state-space dynamics (Gu et al., 2020). Their task performance (e.g., classification accuracy, perplexity) can surpass or compete with transformers, especially for long sequence tasks (Tay et al., 2020). Yet, remarkably, SSMs do not suffer from the quadratic scaling of compute cost of the attention mechanism with context length (Vaswani et al., 2017). Instead, they offer linearly increasing computational costs due to their recurrent formulation. While recurrent neural networks are generally hard to train, SSMs further offer the advantage that they can be implemented as a convolution or as a parallel scan, allowing for easy training on GPUs (Gu et al., 2021a; Smith et al., 2022).

To gain deeper insight on the pertinent internal workings of SSMs, let us examine the original SSM, S4 (Gu et al., 2021a), which captures the essence of the family of subsequent SSM architectures. S4 models can perform their computations using one of three representations, which can be transformed into each other and serve different functional purposes:

$$\dot{\boldsymbol{x}}(t) = \boldsymbol{A}\boldsymbol{x}(t) + \boldsymbol{B}u(t), \qquad\qquad y(t) = \boldsymbol{C}\boldsymbol{x}(t) \tag{1}$$

$$\boldsymbol{x}_k = \overline{\boldsymbol{A}}\boldsymbol{x}_{k-1} + \overline{\boldsymbol{B}}u_k, \qquad\qquad y_k = \overline{\boldsymbol{C}}\boldsymbol{x}_k \tag{2}$$

$$\overline{\boldsymbol{K}} = (\overline{\boldsymbol{CB}}, \overline{\boldsymbol{CAB}}, \cdots, \overline{\boldsymbol{CA}}^{L-1}\overline{\boldsymbol{B}}), \quad (\cdots, y_k, \cdots, y_L) = \overline{\boldsymbol{K}} * (\cdots, u_k, \cdots, x_L) \tag{3}$$

The *continuous recurrent* representation in equation 1 processes continuous 1-D signals $u(t)$ to output signals $y(t)$ via an $N$-dimensional latent space $\boldsymbol{x}(t)$: $u(t) \in \mathbb{R} \rightarrow \boldsymbol{x}(t) \in \mathbb{R}^N \rightarrow y(t) \in \mathbb{R}$. The parameters include the state matrix $A \in \mathbb{C}^{N \times N}$ and the matrices $B \in \mathbb{C}^{N \times 1}$ and $C \in \mathbb{C}^{1 \times N}$. The *discrete recurrent* representation in equation 2 assumes constant step sizes $\Delta$ to transform the matrices $\boldsymbol{A}$, $\boldsymbol{B}$, and $\boldsymbol{C}$ into discrete matrices $\overline{\boldsymbol{A}}$, $\overline{\boldsymbol{B}}$, and $\overline{\boldsymbol{C}}$ and enables autoregressive inference when inputs $u_k$ are presented sequentially. The *convolutional* representation in equation 3 transforms the linear time-invariant SSM in equation 2 into a global convolution, which enables efficient, parallelized training when $L$ data points are available in a batch.

Several hardware-aware adjustments have been applied to SSMs to make them more efficient on GPUs. Most notably, it has been shown that $\boldsymbol{A}$ can be diagonalized with little to no detrimental effect on the algorithmic performance (Gupta et al., 2022), leading to the S4 variant *S4D* (Gu et al., 2022) which we use in our work.

In addition, there have been recent efforts to make sequence modeling architectures compatible with neuromorphic hardware. These efforts include SpikeGPT (Zhu et al., 2023), Spiking SSMs (Shen et al., 2024), Stochastic Spiking SSMs (Bal & Sengupta, 2024), and Spiking-S4 (Du et al., 2024). These works focus on demonstrating how Spiking Neural Networks (SNNs) can increase activation sparsity, which may increase energy efficiency on neuromorphic hardware. However, importantly, these works rely on biologically-inspired Leaky Integrate-and-Fire neurons and binary spikes. This prioritization of biological plausibility can leave underutilized the full gambit of capabilities in modern neuromorphic processors like Loihi 2, such as customizable microcode neurons and graded spikes. Additionally, none of these neuromorphic-compatible SSM efforts include implementations on neuromorphic hardware. The lack of any benchmarked SSMs on actual neuromorphic hardware leaves unknown the efficiency of SSMs on neuromorphic hardware in practice, which we evaluate in this work.

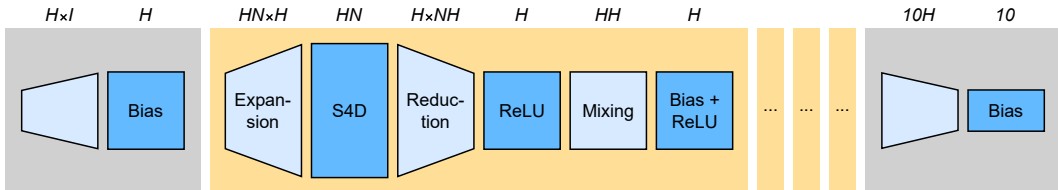

Figure 3: n-S4D model architecture as implemented on Loihi 2. Light blue layers refer to connections and dark blue layers to programmable neurons on Loihi 2. The large yellow box refers to an S4D block, which is repeated four times (represented by the three yellow empty boxes). Variables above each layer denote the dimensionality of the layer, where $H$ denotes the model dimentionality and $N$ denotes the number of hidden states per model dimension.

## 3 NEUROMORPHIC DIAGONAL DEEP STATE-SPACE MODEL

### 3.1 MODEL ARCHITECTURE ON LOIHI 2

Figure 3 shows the SSM model architecture, neuromorphic-S4D (n-S4D), that we implemented on the Loihi 2 neuromorphic processor for sequence classification. n-S4D is inspired by the architecture used by (Gu et al., 2022) with hardware-aware modifications. Our n-S4D network consists of an encoder layer that expands the input to a higher dimensionality, four S4D blocks, and a decoder layer that reduces the dimensionality to the number of output classes. At the top of Figure 3, the dimensionality of each layer of the model is listed, where $I$ represents the input dimensionality, $H$ is the model dimension, and $N$ is the number of hidden states per model dimension; the output dimensionality (number of classes) is 10.

Each S4D block starts with the S4D dynamics implemented as a recurrent network (implementation described in section 3.2), followed by a ReLU activation. After each S4D layer, the dimensions are mixed using a linear projection followed by another ReLU activation. In contrast to the network architecture used to evaluate the original S4D model (Gu et al., 2022), to increase activation sparsity (proportion of zero-valued neural outputs), we only use ReLU activations instead of GLUs and GeLUs. Importantly, when a ReLU neuron outputs a zero, no spike message is sent on Loihi 2; this saves energy thanks to the event-driven nature of Loihi 2. To further simplify the model, we also leave out normalization layers and residual connections.

All S4D layers, ReLU activations, and biases (depicted in dark blue in Figure 3), are implemented as programmable neurons on Loihi 2. All linear projections (depicted in light blue in Figure 3) are implemented in Loihi 2 using linear synapses on neuro cores (weight matrices), including up-projection, expansion, reduction, mixing, and down-projection.

We evaluate two model sizes with 67k parameters ($H = 64, N = 32$) and 265k parameters ($H = 128, N = 64$) for different datasets (see section 4 for details). We optimize how each layer is distributed across neuro cores to achieve uniform compute load; this ensures that no single layer dominates compute time during any given timestep. We place subsequent layers onto neighboring neuro cores to reduce spike Network-on-Chip mesh traffic. These configurations lead to a usage of 31 and 111 neuro cores of a single Loihi 2 chip for the small and large model, respectively.

### 3.2 INCORPORATING SSM DYNAMICS WITHIN PROGRAMMABLE NEURONS

The fact that $\overline{A}$ is diagonal and $\overline{B}$ and $\overline{C}$ are 1-D matrices (see equation 2) implies that each hidden state within the S4D layers evolves independently, without any cross-dependencies with other states. This independence allows for a straightforward implementation of the recurrent SSM dynamics (equation 2) on Loihi 2. Namely, $\overline{B}$ and $\overline{C}$ could be integrated into the synaptic connectivity for expansion and reduction, and $\overline{A}$ could be realized through additional recurrent synaptic connections to the corresponding S4D neuron. However, we opt to modify this straightforward implementation by embedding the complete SSM dynamics directly within the programmable S4D neurons. This approach presents two advantages. Firstly, it minimizes mesh traffic by requiring only a single set of expansion and reduction connections to handle $\overline{B}$ and $\overline{C}$, as opposed to separate connections for

---

**Algorithm 1** Simplified programmed behavior of one instance of the $HN$ S4D Neurons.

1: Initialize constants $a_{\text{real}}$, $a_{\text{imag}}$, $b_{\text{real}}$, $b_{\text{imag}}$, $c_{\text{real}}$, $c_{\text{imag}}$
2: Initialize hidden state $x_{\text{real}}$ and $x_{\text{imag}}$
3: $u \leftarrow \text{RECEIVE\_INPUT}$
4: $x'_{\text{real}} \leftarrow a_{\text{real}} \times x_{\text{real}} - a_{\text{imag}} \times x_{\text{imag}} + b_{\text{real}} \times u$
5: $x_{\text{imag}} \leftarrow a_{\text{real}} \times x_{\text{imag}} - a_{\text{imag}} \times x_{\text{real}} + b_{\text{imag}} \times u$
6: $x_{\text{real}} \leftarrow x'_{\text{real}}$
7: $u \leftarrow 2 \times (c_{\text{real}} \times x_{\text{real}} - c_{\text{imag}} \times x_{\text{imag}})$
8: SEND(y)

---

their real and imaginary components, plus two extra recurrent connections for $\overline{A}$. Secondly, it allows for the use of higher bit precision for the SSM parameters. While standard synaptic weights on Loihi 2 are limited to 8 bits, the states and constants within programmable neurons can be represented with 8 bits, 16 bits, or even 24 bits. This is particularly beneficial for the recurrent weights, which are sensitive to accumulating quantization errors over time.

The described structure leads to the expansion and reduction of synaptic connections through the matrices $\boldsymbol{E}$ and $\boldsymbol{E^T}$, respectively, where $\boldsymbol{E^T} \in \mathbb{R}^{HN \times H}$ is described as follows:

$$\boldsymbol{E}_{i,j} = \begin{cases} 1 & \text{if } N(i-1) + 1 \leq j \leq Ni \\ 0 & \text{otherwise} \end{cases} \tag{4}$$

The matrix $\boldsymbol{E}$ performs the expansion operation, multiplexing the input structure across $N$ parallel paths. The transpose, $\boldsymbol{E^T}$, performs a reduction operation by summing over the $N$ hidden states for each model dimension.

The behavior of a single microcoded S4D neuron is detailed in Algorithm 1. For readability, we omit both bit-shift operations that are necessary due to the mixed precision of different weights, activations, and states and we omit jumps between memory registers. We denote the $HN$ entries of $\overline{A}$ as $a_{\text{real}}$ and $a_{\text{imag}}$ and denote the entries for $\overline{B}$ and $\overline{C}$ analogously. Each S4D layer contains $HN$ individual S4D neurons.

## 3.3 Post Training Quantization and Quantization Aware Fine Tuning

As all computations on Loihi 2 are performed in fixed precision, we describe in this section how we quantize our models for training and inference.

All models are pre-trained in full precision using the convolutional view of n-S4D. After pre-training, we switch to the recurrent mode and quantize our models for inference on Loihi 2 using Post Training Quantization (PTQ) leading to an expected drop of accuracy. To recover the accuracy of the quantized model, we re-train the models using Quantization Aware Fine Tuning (QAFT) in recurrent mode with Loihi 2-specific bit-widths and precisions for one epoch.

All activations are quantized using a bit-width of 24 bits, with 6 to 8 bits being allocated to the fractional part (precision) and the remaining bits for the integer part. While the $\overline{A}$, $\overline{B}$, and $\overline{C}$ matrices of the S4D layers are quantized with a fixed bit-width of 16 bits using a precision of 13 bits, the parameters (weights and biases) of the feed-forward layers such as the encoder, decoder, expansion, reduction and mixing layers are quantized using 8 bits with dynamic precision. In the case of dynamic precision, we calculate scaling factors based on the maximum absolute value of the relevant tensor to use the full dynamic range.

All tensors $\boldsymbol{X}$ are kept in full precision while simulating the effects of quantization ($\hat{\boldsymbol{X}}$) during the forward path:

$$\hat{\boldsymbol{X}} = \lfloor \boldsymbol{X}s \rfloor \hat{d}, \tag{5}$$

where $s$ scales the tensor to the desired precision of $b$ bits and the floor operator $\lfloor \cdot \rfloor$ denotes the truncation of the fractional part. The scaling factor can either be calculated by $s = 2^b$ in the case of a fixed precision or $s = 2^b/|\boldsymbol{X}|_{\text{max}}$ in the case of dynamic scaling. To be fully accurate to Loihi 2's fixed precision arithmetic, we also quantize the descaling factor $d = 1/s$ with a fixed precision of 16 bits. To allow gradient flow in the backward computation, we use a straight-through estimator (Bengio et al., 2013).

Table 1: Comparison against leading reported test accuracies from prior works (Transformer, CNN, RNN, SSM) on the sMNIST, psMNIST, and sCIFAR datasets.

| Model (Input length) | sMNIST (784) | psMNIST (784) | sCIFAR (1024) |
|---|---|---|---|
| Transformer (Vaswani et al., 2017; Trinh et al., 2018) | 98.9 | 97.9 | 62.2 |
| CCNN (Romero et al., 2022) | **99.72** | **98.84** | **93.08** |
| LipschitzRNN (Erichson et al., 2020) | 99.4 | 96.3 | 64.2 |
| LSSL (Gu et al., 2021b) | 99.53 | 98.76 | 84.65 |
| S4 (Gu et al., 2021a; 2022) | 99.63 | 98.70 | 91.80 |
| S4D-LegS (Gu et al., 2022) | - | - | 89.92 |
| Liquid-S4 (Hasani et al., 2022) | - | - | 92.02 |
| S5 (Smith et al., 2022) | 99.65 | 98.67 | 90.10 |
| Q-S5 (8 bit precision PTQ) (Abreu et al., 2024) | 96.27 | - | 44.83 |
| Q-S5 (8 bit precision QAFT) (Abreu et al., 2024) | 99.54 | - | 86.95 |
| AHP SNN on Loihi 1 (Rao et al., 2022) | 96.00 | - | - |
| n-S4D, full precision **(Ours)** | 99.51 | 97.53 | 86.53 |
| n-S4D, after PTQ **(Ours)** | 99.20 | 92.45 | 71.74 |
| **n-S4D, on Loihi 2 after QAFT (Ours)** | 99.20 | 96.16 | 84.13 |

In order to extract the quantized parameters after QAFT or to perform PTQ, we can use equation 5 without the descaling part $\hat{d}$. The descaling factor $\hat{d}$ for the activations is then applied in the microcoded neuron dynamics on Loihi 2. The fake quantization hooks and the switch to the recurrent mode slow the training substantially, hence we apply QAFT for only one epoch.

## 4 RESULTS

We evaluate our n-S4D model running on Loihi 2 and Jetson Orin Nano on the datasets sequential MNIST (sMNIST, LeCun et al., 2010), permuted sequential MNIST (psMNIST, LeCun et al., 2010), and sequential CIFAR10 (sCIFAR, Krizhevsky, 2009).

### 4.1 ACCURACY AND PARAMETER COUNT

Table 1 shows the accuracy on sMNIST, psMNIST, and sCIFAR of our n-S4D model in full precision, after quantization, and on Loihi 2 in comparison to other models. Although we use a simplified version of the S4D model, by only using ReLU activations and no normalization (see section 3.1), the performance in full precision drops by only less than one to four percentage points compared to the more complex S4 and S4D models on all three datasets.

We observe a drop in accuracy when preparing the model for deployment on Loihi 2 by switching to the recurrent mode and quantizing the model after training (PTQ). Precisely, the accuracy only drops substantially on the psMNIST (97.53 % to 92.45 %) and the sCIFAR (86.53 % to 71.74 %) datasets. This drop in accuracy is however less than the drop in accuracy observed when applying PTQ to S5 with 8 bits as reported by Abreu et al. (2024), where the accuracy drops from 99.65 % to 96.27 % for sMNIST and from 90.10 % to 44.83 % for sCIFAR. This loss in accuracy can be recovered to nearly the level of the full precision model by applying QAFT for just one epoch (psMNIST: 96.16 %, sCIFAR: 84.13 %), a similar recovery is observed for QS-5 (Abreu et al., 2024) after 15 epochs of QAFT. Note how there was no switch from the scan mode to the recurrent mode for QS-5 and fake-quantization was applied instead of full quantization, which makes a direct comparison difficult.

Previous non-SSM neuromorphic solutions for sMNIST such as the AHP SNN model on Loihi 1 (Rao et al., 2022) reach a lower accuracy than our n-S4D model on Loihi 2, suggesting a substantial maturing of models and hardware in the neuromorphic domain. Overall, the CCNN model exhibits the highest accuracy on all tasks, while using 2M parameters (Romero et al., 2022). For comparison, our model only uses less than 265k parameters for sCIFAR and 67k parameters for the MNIST datasets.

Table 2: Compute cost comparisons for sample-by-sample and token-by-token based processing. Sample-by-sample based processing assumes that the entire sample (the entire image) is available to the system at the start of processing, whereas token-by-token based processing assumes that tokens (individual pixels) arrive one at a time and are processed sequentially. Implementations use either recurrent (Rec) or convolutional (Conv) n-S4D. Parmater b listed under the column Exec mode indicates batch size. For Loihi 2, "ft" refers to fall-through and "pipe" to pipelined processing.

| | HW | Exec mode | Prec | Acc (↑) (%) | Token-by-Token Processing | | | | Sample-by-Sample Processing | | | |
|---|---|---|---|---|---|---|---|---|---|---|---|---|
| | | | | | Energy (↓) (mJ/token) | Latency (↓) (ms/token) | Throughput (↑) (token/s) | EDP (↓) (μJ s/token) | Energy (↓) (mJ/sample) | Latency (↓) (ms/sample) | Throughput (↑) (sample/s) | EDP (↓) (μJ s/sample) |
| sMNIST | Loihi 2* | Rec (ft) | qint | 99.20 | 0.003 | **0.068** | 14,705 | **0.0002** | 2.678 | 53.314 | 19 | 141.59 |
| | Loihi 2* | Rec (pipe) | qint | 99.20 | **0.002** | 0.168 | **83,343** | 0.0004 | 1.828 | 9.575 | 106 | 17.502 |
| | Jetson† | Rec (b=1) | fp32 | **99.51** | 15.725 | 4.976 | 201 | 79.252 | 12328.652 | 3901.313 | 0.256 | 48.097×10⁶ |
| | Jetson† | Conv (b=1) | fp32 | **99.51** | 23.000 | 6.366 | 157 | 146.418 | 23.000 | **6.366** | 157 | 146.418 |
| | Jetson† | Conv (b=256) | fp32 | **99.51** | - | - | - | - | **0.217** | 8.872 | **28,853** | **1.921** |
| psMNIST | Loihi 2* | Rec (ft) | qint | 96.16 | 0.003 | **0.068** | 14,720 | **0.0002** | 2.678 | 53.262 | 19 | 142.639 |
| | Loihi 2* | Rec (pipe) | qint | 96.16 | **0.002** | 0.168 | **83,349** | 0.0004 | 1.920 | 9.574 | 106 | 15.200 |
| | Jetson† | Rec (b=1) | fp32 | 97.53 | 15.851 | 5.012 | 200 | 70.449 | 12426.807 | 3929.739 | 0.254 | 48.834×10⁶ |
| | Jetson† | Conv (b=1) | fp32 | 97.53 | 23.183 | 6.306 | 158 | 146.187 | 23.183 | **6.306** | 158 | 146.187 |
| | Jetson† | Conv (b=256) | fp32 | 97.53 | - | - | - | - | **0.218** | 8.837 | **28,969** | **1.924** |
| sCIFAR | Loihi 2* | Rec (ft) | qint | 84.13 | 0.016 | **0.066** | 15,259 | **0.0010** | 16.284 | 65.534 | 15 | 1092.808 |
| | Loihi 2* | Rec (pipe) | qint | 84.13 | **0.010** | 0.172 | **81,508** | 0.0017 | 10.355 | 12.735 | 80 | 131.869 |
| | Jetson† | Rec (b=1) | fp32 | **86.53** | 16.106 | 4.978 | 201 | 80.173 | 16492.163 | 5097.42 | 0.194 | 84.067×10⁶ |
| | Jetson† | Conv (b=1) | fp32 | **86.53** | 26.887 | 6.325 | 158 | 170.053 | 26.887 | **6.325** | 158 | 170.053 |
| | Jetson† | Conv (b=64) | fp32 | **86.53** | - | - | - | - | **0.961** | 8.476 | **7,550** | **8.142** |

[*] Loihi 2 workloads were characterized on an Oheo Gulch system with N3C2-revision Loihi 2 chips running on NxCore 2.5.8 and alpha version of the NxKernel API with on-chip IO unthrottled sequencing of input tokens.

[†] GPU workloads were characterized on an NVIDIA Jetson Orin Nano 8GB 15W TDP running Jetpack 5.1.2, TensorRT 8.6.1, Torch-TensorRT 1.3.0. Energy values include CPU_GPU_CV and SOC components as reported by jtop.

[‡] Performance results are based on testing as of September 2024 and may not reflect all publicly available security updates. Results may vary.

## 4.2 COMPUTATIONAL COST

Since our study focuses on a small SSM model appropriate for low-latency and low-power edge processing, we compare our Loihi 2 n-S4D to a recurrent as well as a convolutional implementation of n-S4D on an edge GPU, Nvidia Jetson Orin Nano. While we tried optimizing n-S4D on Jetson Orin Nano, we were unable to create a stronger baseline due to lack of native support for complex numbers in TensorRT (Jeong et al., 2022) for the convolutional implementation (which can not be efficiently implemented using just using just real numbers) and excessively long compilation times for the recurrent model (which can easily be implemented using just real numbers). Consequently, we conducted our assessments using a PyTorch model that had been compiled just-in-time, operating at fp32 precision.

We use the *jtop* API to characterize power. For runtime, only the time spent to input the data and compute the output is considered. For these measurements on Loihi 2, we store a sequence of input values on a neuro core and inject them to the n-S4D network at peak throughput without IO constraints to obtain stable power measurements; this is repeated for 10 representative samples. The energy of the neuro core used to store input values is included in the results for Loihi 2, while the IO power of Jetson is excluded.

The primary point of comparison for the Loihi 2 versus the Jetson implementation is the streaming mode (with a batch size of one) of inference. In we also include the peak-performing batched mode of inference for a representative optimum performance on Jetson. Table 2 reports computational cost of inference on Loihi 2 and Jetson on the three datasets. In addition to energy, latency, and throughput, the energy-delay product (EDP, Shrestha et al., 2024) is reported to more readily compare systems running at different speeds.

**Sample-by-sample processing** The right side of Table 2 shows results from sample-by-sample processing, which assumes that all tokens of a sample are available to the system at the beginning of processing and only a single classification is required for the whole sample. Recurrent formulations

of the model have to process each token sequentially, while convolutional formulations can process the entire sample with a single convolution.

In the online processing regime with a batch size of one, it is evident that Loihi 2 is very efficient compared to the recurrent mode on Jetson and shows better energy per sample of 1.8 mJ compared to 23 mJ for Jetson in the convolutional mode that is favorable for GPU architectures. The throughput and latency on Loihi 2 and Jetson in convolutional mode are also competitive. Jetson, however, achieves peak performance in higher batch mode with substantially reduced energy per sample of 0.22 mJ and 0.96 mJ for sMNIST and sCIFAR along with orders of magnitude higher throughput, which is expected for GPU architectures.

**Token-by-token processing** The middle columns of Table 2 show results of token-by-token processing. This assumes streaming input that requires a classification for every token. For these types of tasks, both Loihi 2 and the recurrent mode on Jetson process all tokens sequentially. The convolution mode on Jetson, however, has to perform a convolution over the entire sequence with every new token. Its latency, energy, throughput, and EDP are therefore the same for processing one token in token-by-token processing as they are for processing one sample in the sample-by-sample processing mode when using the convolutional implementation.

For token-by-token processing, Loihi 2 outperforms Jetson in all metrics on all datasets. Latency and energy are two to three orders of magnitude lower for Loihi 2. This is reflected in EDP: For MNIST workloads, EDP for Loihi 2 is $0.0002\,\mu\mathrm{J}\,\mathrm{s}$ and $0.001\,\mu\mathrm{J}\,\mathrm{s}$ for sCIFAR. In contrast, the recurrent processing on Jetson incurs EDP of $70\,\mu\mathrm{J}\,\mathrm{s}$ and $80\,\mu\mathrm{J}\,\mathrm{s}$ on the respective datasets, demonstrating the efficiency of Loihi 2 in token-by-token processing.

**Fall-through vs. pipelined processing** Table 2 shows the tradeoff between latency and throughput when executing the model on Loihi 2 in fall-through or pipelined processing (see section 2.1). With fall-through processing, we see a latency of $68\,\mu\mathrm{s}$ per token on the sMNIST dataset, compared to $168\,\mu\mathrm{s}$ in pipelined processing. This comes at the cost of throughput of only 14.705 token/s, compared to 83.343 token/s in pipelined processing. The lower throughput per token in fall-through mode results in a higher latency per sample of 53.314 ms compared to the pipelined mode with 9.57 ms. Results on the other datasets highlight the same tradeoff.

## 5 DISCUSSION

Our benchmark results (see section 4) show that the n-S4D model on Loihi 2 particularly excels in online token-by-token inference. Online token-by-token inference is widely applicable in streaming scenarios in which an incoming data stream must be rapidly processed on a token-by-token basis. In the token-by-token scenario, we demonstrate that Loihi 2 can meaningfully leverage its substantively differentiated compute and memory co-located architecture to outperform the baseline Jetson GPU. On sCIFAR, the largest workload we measured, Loihi 2 consumes approximately 1000x less energy with a 75x lower latency and a 75x higher throughput compared to the recurrent implementation of n-S4D on the Jetson GPU. We also benchmark the offline sample-by-sample and batched scenarios, for which we find the Jetson GPU to be preferable to Loihi 2. Our results corroborate the notion that GPU architectures are optimized for offline processing of large amounts of data in parallel. It should be noted that our implementation of n-S4D on Jetson is not completely optimized for speed and efficiency. Creating a stronger baseline implementation on Jetson as well as considering other types of hardware would be valuable future work.

Taken comprehensively, our results provide the first benchmarks of an SSM on a neuromorphic hardware platform versus an edge GPU, comparing both the recurrent and convolution modes and revealing the differences in energy, latency, throughput, and task accuracy. To the best of our knowledge, this is the most holistic picture to date of the merits of neuromorphic hardware for SSM efficiency. Furthermore, by virtue of our focus on SSMs—a family of promising and broadly applicable deep learning sequence models—we build an exemplar for others to replicate and expand upon to help bridge deep learning technology to highly-differentiated computer architectures with compute and memory co-location for substantively improved efficiency.

Promising future work includes the following. The modest drop in accuracy of n-S4D on Loihi 2 in this work could potentially be ameliorated by applying QAFT for a longer duration than one epoch.

The balance of latency, energy, and throughput on Loihi 2 over the continuum between fall-through and pipelined processing could be more extensively characterized; indeed, only the endpoints of this continuum are characterized in this work. Direct extensions of S4, for instance Liquid-S4 (Hasani et al., 2022) or S5 (Smith et al., 2022), could be investigated; these extensions have shown state-of-the-art performance on sequence modeling tasks and are also compatible with Loihi 2. Finally, our work and potential optimizations and extensions can be applied and tested in real-world streaming use-cases, such as keyword-spotting, audio denoising, vision for drone control, autonomous driving, and other latency or energy constrained domains.

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
