# OpenReview forum: "A Diagonal Structured State Space Model on Loihi 2 for Efficient Streaming Sequence Processing"
_ICLR.cc/2025/Conference — ICLR 2025 Conference Withdrawn Submission_

### Official Review · Reviewer_sPCg · 2024-11-02

**Soundness:** 3
**Presentation:** 3
**Contribution:** 2
**Rating:** 5
**Confidence:** 4

**Summary:**

In this paper, the authors demonstrate significant energy savings when running a state-space model in an online setting on the neuromorphic Intel Loihi 2 chip compared to the Jetson Orin Nano GPU. During recurrent-mode inference, the neuromorphic approach achieves markedly improved latency and energy efficiency with minimal performance degradation relative to the full-precision model on the GPU.

**Strengths:**

1. This paper is the first to implement a simplified S4 model on the Intel Loihi 2 neuromorphic chip, utilizing its architecture for efficient online inference.

2. The paper empirically validates that using neuromorphic hardware outperforms GPUs during online execution of S4 models where as the  in an offline setting (i.e. convolution-based operation of S4) GPUs fare better than neuromorphic chips.

**Weaknesses:**

1. While this work presents engineering value, its algorithmic contributions lack substantial novelty. The authors introduce a neuromorphic variant of the S4 model without normalization layers and residual connections. Although omitting these layers may not lead to significant accuracy drops on simpler datasets, such as s-MNIST and s-CIFAR, this approach may not scale effectively to more complex tasks requiring long-range dependency modeling

2. While its exciting that authors have directly reported results after running their model on Intel Loihi 2 chip, the empirical evaluation can be more rigorous. Instead of just MNIST, CIFAR the authors can evaluate running their model on the long range arena benchmark or Speech Command dataset. This is to better capture the long-range dependency capturing capability of the proposed architecture.

**Questions:**

1. Can the authors scale the model to datasets such as long range arena benchmark, speech command dataset? This will demonstrate the scalability of the approach on more complex long-range dependency tasks.

2. Why was Jetson Orin Nano chosen specifically as the GPU?

Please check weaknesses as well.

---

### Official Review · Reviewer_zWMw · 2024-11-02

**Soundness:** 2
**Presentation:** 4
**Contribution:** 2
**Rating:** 3
**Confidence:** 4

**Summary:**

The paper implements a diagonal state-space model (SSM) variant, S4D, on the neuromorphic chip Loihi 2. This chip was designed for spiking neural network inference. Since spiking neural networks follow linear dynamical systems while membrane potentials are below the spike threshold, Loihi 2 appears to be well suited for SSM inference. The paper discusses their mapping strategy including partitioning of operations to cores and quantization methods. Task performance is compared between the full precision software implementation, post training quantization and a quantization aware fine-tuned model deployed on Loihi 2. It seems that a bare-metal implementation of an 8-bit model on Loihi 2 is compared against a JIT compiled full precision pytorch model deployed on a Jetson Orin Nano in terms of energy and latency.

**Strengths:**

The writing is concise and clear. It introduces all relevant concepts to convey their methodology to the reviewer (who is familiar with the SSM literature, spiking neural network literature and deployment on neuromorphic platforms). The reviewer further appreciates the authors transparency in terms of their methods limitations (e.g. not optimizing for TensorRT).

The topic of the paper is of due interest to the machine learning community. State-space models are increasingly popular and their deployment efficiency is of interest to machine learning practitioners. In this context, the paper shows a successful deployment of a particular SSM, namely S4D, on an efficient inference chip.

Spiking neural networks implement linear dynamical systems for their sub-threshold activity. It is hence expected that there exists a naive mapping of SSMs to neuromorphic chips that support graded spikes (network packages that have a payload of multiple bit). The paper however provides a hardware aware implementation that by the means of system efficiency considerations such as minimizing communication sounds well designed. The provided material is however not detailed enough to fully review this central contribution.

**Weaknesses:**

Despite the thorough and clear documentation of the work, the reviewer has to list the following concerns that underline the assigned score for this submission.
1. ICLR might not be the appropriate target for this work. The major contribution of the paper is the successful deployment of S4D on the Loihi 2 chip. With this implementation centric work, the  paper might be better placed at a systems conference where much more space can be attributed to a detailed description of this core contribution (MLSys, AICAS etc). The algorithmic contributions are very incremental in terms of model architecture (replacing activation functions) and quantization.
   Why is the deployment the core contribution? Quantization methods for SSMs were investigated in [1.], and 8-bit quantization is an established standard in the ML community with most deployment frameworks supporting 8-bit precision at least for matmuls. The A, B, C matrices of SSMs might be a different matter, but does not provide data or comparison against established methods and references [1.] for this sake. At the same time, the task evaluation on sequential MNIST and CIFAR-10 is quite weak to claim contributions towards quantization methods. E.g. line 361f highlights that the accuracy drop introduced by 8-bit quantization is very small compared to the baseline. This is not a surprise on small scale datasets. We know for example from language models that naive quantization aware training works well up to a certain scale around 7B parameters and only breaks afterwards.
2. The hardware aware implementation is a strength of the paper. Unfortunately, a comparison to the obvious naive implementation that the authors discuss as well is missing. This would give readers a clearer picture of the significance of the hardware aware implementation. Without this data added, it is not clear that the proposed implementation is actually a significant contribution. A comparison to existing spiking neural network implementations on Loihi 2 would further add value to the paper.
3. The hardware-aware implementation on Loihi 2 is evaluated against a JIT compiled full precision pytorch model deployed on a Jetson Orin Nano. There are a couple of issues with this comparison.
	- Integer precision for matrix operations poses a significant reduction in energy spend on computational operations and memory movement. Hence, comparing a quantized model against a full precision model is not fair.
	- The implementation on Loihi 2 is tuned towards the system architecture of the Loihi system, while the authors mention that the Jetson implementation was just-in-time compiled from a torch model. It is not clear from the presentation of the paper, which optimizations were leveraged by the JIT compiler. The authors are transparent about this issue and even point out that they were not able to use NVIDIA's TensorRT framework for efficient deployment. It is hence likely that a highly optimized Loihi 2 implementation is compared to a poorly optimized CUDA implementation on the Jetson Orin Nano. To strengthen the paper, it is recommended to provide an implementation of similar sophistication or at least provide insights into which optimizations were part of the compilation procedure.
	- The Jetson Orin Nano is oversized for the networks implemented. An Orin Nano has 512 to 1024 CUDA cores plus 3rd generation tensor cores as well as 4 to 8 GB DRAM. Running networks with up to 275.000 parameters favors the smaller Loihi 2 chip with only 128 cores and no DRAM. It is not clear from the paper if the Jetson was fully utilized. For example the power of the large DRAM might significantly contribute to the energy consumption of the Jetson despite not requiring DRAM at all for such small models. The results in table 2 suggest that only the large batch size of 64 could fully utilize the Jetson system - and in this setting the Jetson outperforms the Loihi 2 implementation. To allow for a fair comparison, it would be valuable to add results of an optimized implementation on a small deep learning accelerator of similar size as the Loihi 2 system. For example the Hailo-8 M.2 chip might be a better system for comparison than the oversized Jetson system. Another alternative to strengthening the results would be to run larger networks that saturate the compute and memory capacity of the Jetson. Perhaps the comparison with a low-power CPU with sufficiently large cache to host the 275.000 kb for the parameters might be a fairer comparison than the Jetson.
	- To the best of the knowledge of the reviewer, there is no other implementation of SSMs on inference hardware that reports energy or latency numbers. In this environment, it would be valuable to add related works that optimize implementations of related recurrent neural networks for example on FPGAs. This would contribute to setting the present paper in the context of the RNN inference landscape.

[1. Q-S5: Towards Quantized State Space Models](https://arxiv.org/abs/2406.09477)

**Questions:**

- The paper implicitly raises an important question, which is of high interest to the neuromorphic computing community: Are ReLU activated SSM units sufficient to serve neuromorphic platforms compared to bio-plausible spiking neural networks. It would be very interesting to compare the sparsity that this work obtains versus to the sparsity generated by standard SNNs such as (adaptive) leaky integrate-and-fire neurons. Can the authors share insights into the sparsity of their models after ReLU?
- How does the energy consumption of the system change with activation sparsity? A figure with sparsity on the x-axis and EDP on the y-axis would be great. It's clear that the sparsity depends on the task and learning method. Yet, the requested data could be collected merely from a random network with varying bias before the ReLU.
- What precisely is meant in line 422 by IO. Does this take all the memory movement of the DRAM into account (DRAM I/O) or just the one-off loading of weights into GPU memory plus loading the data into GPU memory?

**Details Of Ethics Concerns:**

The restrictive access to the Loihi 2 platform might favor bias towards this system. Although this is hidden from the reviewer, it is quite likely that the authors are associated with Intel, the company behind the Loihi 2 system. A fair comparison becomes even more important in this case. As discussed in the "weaknesses" section, the reviewer is not convinced that a fair comparison to other systems was conducted both in terms of hardware and software at the level that would be expected from a top ML conference.

---

### Official Review · Reviewer_ssK5 · 2024-11-04

**Soundness:** 3
**Presentation:** 2
**Contribution:** 2
**Rating:** 5
**Confidence:** 4

**Summary:**

This paper implements a neuromorphic Diagonal Deep State Space Model (n-S4D) on Loihi 2, a neuromorphic chip, and applies post-training quantization along with quantization-aware fine-tuning to mitigate performance degradation. The n-S4D model achieves competitive results on several sequential datasets compared to existing models. Additionally, the authors highlight the advantages of the n-S4D model in terms of energy consumption, latency, throughput, and Energy-Delay Product (EDP) when processing tokens sequentially.

**Strengths:**

- This is the first paper to implement a spiking version of SSM models on neuromorphic hardware and achieve comparable results.
- The authors apply different quantization precisions for various components of the n-S4D model, effectively reducing quantization errors.
- This paper provides extensive system metrics on the hardware, showcasing the actual performance and computational cost of the spiking models. Notably, for token-by-token processing, the advantages in energy consumption, latency, throughput, and EDP are compelling.

**Weaknesses:**

- The primary modification to the model structure involves using ReLU instead of GLUs/GeLUs as the activation function to increase activation sparsity. It would be better if the authors could visualize the activation maps to substantiate this claim.

- The paper notes that different bit-widths were chosen for activations, synaptic weights, and the A, B, C matrices of the S4D layers. Including an ablation study on the bit-width selection would strengthen the paper by illustrating the trade-off between accuracy and efficiency.

- The generalizability of this approach is limited. Showing results for the spiking version of other SSM models, such as Mamba or Mamba2, would enhance the paper’s contribution.

**Questions:**

Same as the weaknesses.

---

### Official Review · Reviewer_aLHr · 2024-11-05

**Soundness:** 3
**Presentation:** 2
**Contribution:** 1
**Rating:** 3
**Confidence:** 4

**Summary:**

This paper proposes a mapping scheme of state space models on Loihi 2, and demonstrates significant energy efficiency and latency advantages compared to full-precision models. The implementation is tailored towards Loihi-specific features and constraints, and is operated in online mode as a recurrent network.

**Strengths:**

1. First evaluation of SNNs on neuromorphic hardware, and comprehensive comparison with GPUs.

2. Important research direction as the need for efficient SSMs grow.

**Weaknesses:**

1. I do not see a lot of technical novelty in this work. The SSM models and the quantization techniques are all well-explored in the community. Though the authors evaluate the SSMs on Loihi, I do not see any significant challenges and novel solutions to address the Loihi-specific constraints.

2. The evaluation benchmarks adopted are too simple and outdated. The authors should report results on long range modeling benchmarks, such as Long range Arena and other modern NLP benchmarks.

3. SSMs can be mapped to programmable spiking neurons as shown in [1]. The comparison with this work should be included.

[1] https://arxiv.org/abs/2406.02923v1

**Questions:**

Please see above.

---

### Note · Authors · 2024-11-30

**Comment:**

Dear Reviewers,

Thank you for your thoughtful and detailed feedback on our submission. We greatly appreciate the time and effort you put into reviewing our work and providing valuable insights.

We are pleased that you found the writing clear and the topic relevant. Your suggestions, particularly on system comparisons, sparsity analysis, and the contextual framing of contributions, are deeply appreciated and will guide us in refining the work.

After careful consideration, we have decided to withdraw our submission. We acknowledge your suggestion that this work might be better suited for a systems-oriented venue where there is more scope to elaborate on implementation details.

Thank you once again for your constructive feedback. We look forward to incorporating these suggestions as we further develop this work.

**Withdrawal Confirmation:**

I have read and agree with the venue's withdrawal policy on behalf of myself and my co-authors.